# NON-BACKTRACKING GRAPH NEURAL NETWORK

## ABSTRACT

The celebrated message-passing updates for graph neural networks allow the representation of large-scale graphs with local and computationally tractable updates. However, the local updates suffer from backtracking, i.e., a message flows through the same edge twice and revisits the previously visited node. Since the number of message flows increases exponentially with the number of updates, the redundancy in local updates prevents the graph neural network from accurately recognizing a particular message flow for downstream tasks. In this work, we propose to resolve such a redundancy via the non-backtracking graph neural network (NBA-GNN) that updates a message without incorporating the message from the previously visited node. We further investigate how NBA-GNN alleviates the over-squashing of GNNs, and establish a connection between NBA-GNN and the impressive performance of non-backtracking updates for stochastic block model recovery. We empirically verify the effectiveness of our NBA-GNN on long-range graph benchmark and transductive node classification problems.

## 1 INTRODUCTION

Recently, graph neural networks (GNNs) (Kipf & Welling, 2016; Hamilton et al., 2017; Xu et al., 2019) have shown great success in various applications, including but not limited to, molecular property prediction (Gilmer et al., 2017) and community detection (Bruna & Li, 2017). Such success can be largely attributed to the message-passing structure of GNNs, which provides a computationally tractable way of incorporating the overall graph through iterative updates based on local neighborhoods. However, the message-passing structure also brings challenges due to the parallel updates and memory-less behavior of messages passed along the graph.

Figure 1: Message flows of simple (above) and non-backtracking (below) updates.

In particular, the message flow in a GNN is prone to backtracking, where the message from vertex $i$ to vertex $j$ is reincorporated in the subsequent message from $j$ to $i$, e.g., Figure 1. Since the message-passing iteratively aggregates the information, the GNN inevitably encounters an exponential surge in the number of message flows, proportionate to the vertex degrees. This issue is compounded by backtracking, which accelerates the growth of message flows with redundant information.

Interestingly, despite these challenges, non-backtracking updates—a potential solution—have been largely overlooked in the existing GNN research, while they have been thoroughly investigated for non-GNN message-passing algorithms or random walks (Fitzner & van der Hofstad, 2013; Rappaport et al., 2017) (Figure 1). For example, given a pair of vertices $i, j$, the belief propagation algorithm (Pearl, 1982) forbids an $i \rightarrow j$ message from incorporating the $j \rightarrow i$ message. Another example is the non-backtracking random walks (Alon et al., 2007) which are non-Markovian walks that do not traverse the same edge twice and revisit the previous node. Such classic algorithms have demonstrated great success in applications like probabilistic graphical model inference and stochastic block models (Massoulié, 2014; Bordenave et al., 2015; Abbe & Sandon, 2015b). In particular, the spectrum of the non-backtracking operator contains more useful information than that of the adjacency matrix in revealing the hidden structure of a graph model (Bordenave et al., 2015).

**Contribution.** In this work, we propose the non-backtracking graph neural network (NBA-GNN) which employs non-backtracking updates on the messages, i.e., forbids the message from vertex $i$ to vertex $j$ from being incorporated in the message from vertex $j$ to $i$. To this end, we associate the hidden features with transitions between a pair of vertices, e.g., $h_{j \to i}$, and update them from features associated with non-backtracking transitions, e.g., $h_{k \to j}$ for $k \neq i$.

To motivate our work, we formulate "message flows" as the sensitivity of a GNN with respect to walks in the graph. Then we explain how the message flows are redundant; the GNN's sensitivity of a walk with backtracking transitions can be covered by that of other non-backtracking walks. We explain how the redundancy is harmful to the GNN since the number of walks increases exponentially as the number of layers grows and the GNN becomes insensitive to a particular walk information. Hence, reducing the redundancy by only considering non-backtracking walks would benefit the message-passing updates to better recognize each walk's information. We further make a connection from our sensitivity analysis to the over-squashing phenomenon for GNNs (Topping et al., 2022; Black et al., 2023; Di Giovanni et al., 2023) in terms of access time.

Furthermore, we analyze our NBA-GNNs from the perspective of over-squashing and their expressive capability to recover sparse stochastic block models (SBMs). To this end, we prove that NBA-GNN improves the Jacobian-based measure of over-squashing (Topping et al., 2022) compared to its original GNN counterpart.

Next, we investigate NBA-GNN's proficiency in node classification within SBMs and its ability to distinguish between graphs originating from the Erdős–Rényi model or the SBM, from the results of (Stephan & Massoulié, 2022; Bordenave et al., 2015). Unlike traditional GNNs that operate on adjacency matrices and necessitate an average degree of at least $\Omega(\log n)$, NBA-GNN demonstrates the ability to perform node classification with a substantially lower average degree bound of $\omega(1)$ and $n^{o(1)}$. Furthermore, the algorithm can accurately classify graphs even when the average degree remains constant.

Finally, we empirically evaluate our NBA-GNN on the long-range graph benchmark (Dwivedi et al., 2022) and transductive node classification problems (Sen et al., 2008; Pei et al., 2019). We observe that our NBA-GNN demonstrates competitive performance and even achieves state-of-the-art performance on the long-rage graph benchmark. For the node classification tasks, we demonstrate that NBA-GNN consistently improves over its conventional GNN counterpart.

To summarize, our contributions are as follows:

- We propose NBA-GNN as a solution for the message flow redundancy problem in GNNs.
- We analyze how the NBA-GNN alleviates over-squashing and is expressive enough to recover sparse stochastic block models with an average degree of $o(\log n)$.
- We empirically verify our NBA-GNNs to show state-of-the-art performance on the long-range graph benchmark and consistently improve over the conventional GNNs across various tasks.

## 2 RELATED WORKS

**Non-backtracking algorithms.** Non-backtracking updates have been considered by many classical algorithms (Newman, 2013; Kempton, 2016). Belief propagation (Pearl, 1982) infers the marginal distribution on probabilistic graphical models, and has demonstrated success for tree graphs (Kim & Pearl, 1983) and graphs with large girth (Murphy et al., 2013). Moreover, Mahé et al. (2004) and Aziz et al. (2013) suggest graph kernels between labeled graphs utilizing non-backtracking walks, and Krzakala et al. (2013) has first used it for node classification. Furthermore, the non-backtracking has been shown to yield better spectral separation properties, and its eigenspace contains information about the hidden structure of a graph model (Bordenave et al., 2015; Stephan & Massoulié, 2022).

Additionally, Chen et al. (2017) has first used the non-backtracking operator in GNNs, though limited to community detection tasks. Chen et al. (2022) has removed redundancy by computing non-redundant tree for every nodes, inevitably suffering high complexity. We highlight the differences between the related works and our NBA-GNN in Appendix A, due to limited space.

**Analyzing over-squashing of GNNs.** When a node receives information from a $k$-hop neighbor node, an exponential number of messages passes through node representations with fixed-sized vec-

tors. This leads to the loss of information denoted as *over-squashing* (Alon & Yahav, 2020), and has been formalized in terms of sensitivity (Topping et al., 2022; Di Giovanni et al., 2023). Hence, sensitivity is defined as the Jacobian of a node feature at a GNN layer with respect to the input node, and can be upper bounded via the graph topology. Stemming from this, graph rewiring methods mitigate over-squashing by adding or removing edges for an optimal graph topology to compute (Topping et al., 2022; Black et al., 2023; Di Giovanni et al., 2023). Another line of work uses global aspects, e.g., connecting a virtual global node (Cai et al., 2023) or using graph Transformers (Ying et al., 2021; Kreuzer et al., 2021; Rampášek et al., 2022; Shirzad et al., 2023).

**Expressivity of GNNs with the Weisfeiler-Lehman (WL) test.** In the past few years, researchers have conducted significant studies on various aspects of the expressive power of GNNs. One line of research involves comparing GNNs with the WL test (Leman & Weisfeiler, 1968) to assess their expressiveness. For instance, Xu et al. (2019) demonstrated that MPNNs are at best as powerful as the WL test and introduced the graph isomorphism network (GIN), which matches the representational power of the WL test. Besides, Kanatsoulis & Ribeiro (2023) analyze the expressivity of GNNs utilizing linear algebraic tools and eigenvalue decomposition of graph operators.

**Expressivity of GNNs for the stochastic block model (SBM).** Furthermore, certain studies have analyzed the expressive power of GNNs using variations of the SBM (Holland et al., 1983). Fountoulakis et al. (2022) established conditions for the existence of graph attention networks (GATs) that can precisely classify nodes in the contextual stochastic block model (CSBM) with high probability. Similarly, Baranwal et al. (2022) investigated the effects of graph convolutions within a network on the XOR-CSBM. These works focused primarily on the probability distribution of node features, such as the distance between the means of feature vectors.

## 3 NON-BACKTRACKING GRAPH NEURAL NETWORK

### 3.1 MOTIVATION FROM SENSITIVITY ANALYSIS

We first explain how the conventional message-passing updates are prone to backtracking. To this end, consider a simple, undirected graph $\mathcal{G} = (\mathcal{V}, \mathcal{E})$ where $\mathcal{N}(i)$ denotes the neighbor of the node $i$. Each node $i \in \mathcal{V}$ is associated with a feature $x_i$. Then the conventional graph neural networks (GNNs), i.e., message-passing neural networks (MPNNs) (Gilmer et al., 2017), iteratively update the $t$-th layer node-wise hidden feature $h_i^{(t)}$ as follows:

$$h_i^{(t+1)} = \phi^{(t)}\left(h_i^{(t)}, \left\{\psi^{(t)}\left(h_i^{(t)}, h_j^{(t)}\right) : j \in \mathcal{N}(i)\right\}\right), \tag{1}$$

where $\phi^{(t)}$ and $\psi^{(t)}$ are architecture-specific non-linear update and permutation invariant aggregation functions, respectively. Our key observation is that the message from the node feature $h_i^{(t)}$ to the node feature $h_j^{(t+1)}$ is reincorporated in the node feature $h_i^{(t+2)}$, e.g., Figure 3a shows the computation graph of conventional GNNs where redundant messages are incorporated.

**Sensitivity analysis.** To concretely describe the backtracking nature of message-passing updates, we formulate the sensitivity of the final node feature $h_i^{(T)}$ with respect to the input as follows:

$$\sum_{j \in \mathcal{V}} \frac{\partial h_i^{(T)}}{\partial h_j^{(0)}} = \sum_{s \in \mathcal{W}(i)} \prod_{t=1}^{T} \frac{\partial h_{s(t)}^{(t)}}{\partial h_{s(t-1)}^{(t-1)}}, \tag{2}$$

where $h_i^{(0)} = x_i$, $\mathcal{W}(i)$ denotes the set of $T$-step walks ending at node $i$, and $s(t)$ denotes the $t$-th node in the walk $s \in \mathcal{W}(i)$. Intuitively, this equation shows that a GNN with $T$ layer recognizes the graph via aggregation of random walks with length $T$. Our key observation from Equation (2) is on how the feature $h_i^{(T)}$ is insensitive to the information to an initial node feature $h_j^{(0)}$, due to the information being "squashed" by the aggregation over the exponential number of walks $\mathcal{W}(i)$. A similar analysis has been conducted on how a node feature $h_i^{(T)}$ is insensitive to the far-away initial node feature $h_j^{(0)} = x_j$, i.e., the over-squashing phenomenon of GNNs (Topping et al., 2022).

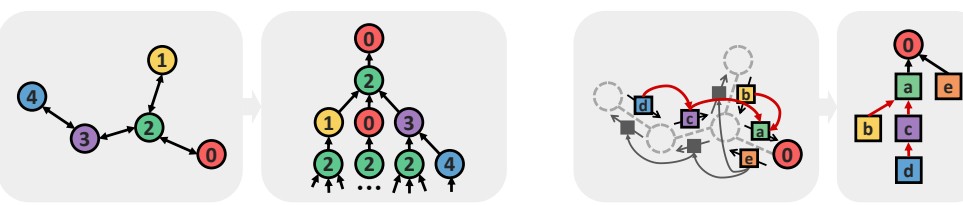

(a) Computation graph of typical GNNs      (b) Computation graph of NBA-GNNs

Figure 3: Computation graph of typical GNN and NBA-GNN predicting node "0". (a) Redundant messages increase the size of the computation graph. (b) NBA-GNN assigns a pair of features for each edge and update them via non-backtracking message passing, reducing redundant messages.

**Redundancy of walks with backtracking.** In particular, a walk $s$ randomly sampled from $\mathcal{W}(i)$ is likely to contain a transition that backtracks, i.e., $s(t) = s(t + 2)$ for some $t < T$. Then the walk $s$ would be redundant since the information is contained in two other walks in $\mathcal{W}(i)$: $s(0), \ldots, s(t + 1)$ and $s(0), \ldots, s(t + 1), s(t) = s(t+2), s(t+3), ..., s(T)$. See Figure 2 for an illustration. This idea leads to the conclusion that non-backtracking walks, i.e., walks that do not contain backtracking transitions, are sufficient to express the information in the walks $\mathcal{W}(i)$. Since the exponential number of walks in $\mathcal{W}(i)$ causes the GNN to be insensitive to a particular walk information, it makes sense to design a non-

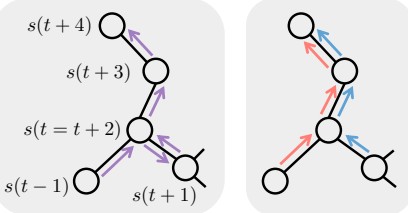

Figure 2: Two non-backtracking walks (right) are sufficient to express information in a walk with backtracking transition.

backtracking GNN that is sensitive to the constrained set of non-backtracking walks. We note that similar concepts of redundancy for GNNs have been studied in prior works (Jia et al., 2020; Chen et al., 2022). See Appendix A for a detailed comparison.

**Relation to over-squashing.** Finally, we point out an intriguing motivation for our work in terms of over-squashing. In particular, we note that Di Giovanni et al. (2023) analyzed the lower bound for the Jacobian obstruction that measures the degree of over-squashing in terms of access time with respect to a simple random walk. They conclude that the degree of over-squashing, i.e., the size of Jacobian obstruction, is higher for a pair of nodes with longer access time.

Hence, to design a GNN architecture robust to the over-squashing phenomenon, one could (i) propose a random walk that has shorter access time in general for a pair of nodes in the graph, and (ii) design a GNN that aligns with the optimized random walk. Non-backtracking random walks have been empirically shown and believed to generally yield faster access time than simple random walks (Lin & Zhang, 2019; Fasino et al., 2023). Therefore, one could aim to design a GNN architecture that aligns with the non-backtracking random walks.

However, to the best of our knowledge, there is no formal proof of scenarios where non-backtracking random walks yield a shorter access time. As a motivating example, we provide a theoretical result comparing the access time of non-backtracking and simple random walks for tree graphs.

**Proposition 1.** *Given a tree $\mathcal{G} = (\mathcal{V}, \mathcal{E})$ and a pair of nodes $i, j \in \mathcal{V}$, the access time of begrudgingly backtracking random walk is equal or smaller than that of a simple random walk, where the equality holds if and only if the random walk length is 1.*

The begrudgingly backtracking random walk (Rappaport et al., 2017) modifies non-backtracking random walks to remove "dead ends" for tree graphs. Please refer to Appendix B for the proof.

### 3.2 Method Description

In this section, we present our **N**on-**BA**cktracking GNNs (NBA-GNN) with the motivation described in Section 3.1. Given an undirected graph $\mathcal{G} = (\mathcal{V}, \mathcal{E})$, our NBA-GNN associates a pair of hidden features $h_{i \to j}^{(t)}, h_{j \to i}^{(t)}$ for each edge $\{i, j\}$. Then the non-backtracking message passing update for a

hidden feature $h_{j \to i}^{(t)}$ is defined as follows:

$$h_{j \to i}^{(t+1)} = \phi^{(t)}\left(h_{j \to i}^{(t)}, \left\{\psi^{(t)}\left(h_{k \to j}^{(t)}, h_{j \to i}^{(t)}\right) : k \in \mathcal{N}(j) \setminus \{i\}\right\}\right), \tag{3}$$

where $\phi^{(t)}$ and $\psi^{(t)}$ are backbone-specific non-linear update and permutation-invariant aggregation functions at the $t$-th layer, respectively. For example, $\psi^{(t)}$ and $\phi^{(t)}$ are multi-layer perceptron and summation over a set for the graph isomorphism network (Xu et al., 2019), respectively. Given the update in Equation (3), one can observe that the message $h_{i \to j}^{(t)}$ is never incorporated in the message $h_{j \to i}^{(t+1)}$, and hence the update is free from backtracking.

**Initialization and node-wise aggregation of messages.** The message at the 0-th layer $h_{i \to j}^{(0)}$ is initialized by encoding the node features $x_i, x_j$, and the edge feature $e_{ij}$ using a non-linear function $\phi$. After updating hidden features for each edge based on Equation (3), we apply permutation-invariant pooling over all the messages for graph-wise predictions. Since we use hidden features for each edge, we construct the node-wise predictions at the final layer as follows:

$$h_i = \sigma\left(\rho\left\{h_{j \to i}^{(T)} : j \in \mathcal{N}(i)\right\}, \rho\left\{h_{i \to j}^{(T)} : j \in \mathcal{N}(i)\right\}\right), \tag{4}$$

where $\sigma$ is a non-linear aggregation function with different weights for incoming edges $j \to i$ and outgoing edges $i \to j$, $\rho$ is a non-linear aggregation function invariant to the permutation of nodes in $\mathcal{N}(i)$. We provide a computation graph of NBA-GNNs in Figure 3b to summarize our algorithm.

**Begrudgingly backtracking update.** While the messages from our update are resistant to backtracking, a message $h_{j \to i}^{(t)}$ may get trapped in node $i$ for the special case when $\mathcal{N}(i) = \{j\}$. To resolve this issue, we introduce a simple trick coined begrudgingly backtracking update (Rappaport et al., 2017) that updates $h_{i \to j}^{(t+1)}$ using $h_{j \to i}^{(t)}$ only when $\mathcal{N}(i) = \{j\}$. We empirically verify the effectiveness of begrudgingly backtracking updates in Section 5.3.

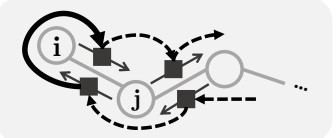

Figure 4: Begrudgingly backtracking update (solid).

**Implementation.** To better understand our NBA-GNN, we provide an example of non-backtracking message-passing updates with a GCN backbone (Kipf & Welling, 2016), coined NBA-GCN. The message update at the $t$-th layer of NBA-GCN can be written as follows:

$$h_{j \to i}^{(t+1)} = h_{j \to i}^{(t)} + \sigma_{\text{GCN}}\left(\frac{1}{|\mathcal{N}(j)| - 1}\mathbf{W}^{(t)}\sum_{k \in \mathcal{N}(j) \setminus \{i\}} h_{k \to j}^{(t)}\right), \tag{5}$$

where $\sigma_{\text{GCN}}$ is an element-wise nonlinear function, e.g., rectified linear unit (Agarap, 2018), $\mathbf{W}^{(t)}$ is the weight matrix, and the messages are normalized by their population $|\mathcal{N}(j)| - 1$.

## 4 THEORETICAL ANALYSIS

In this section, we analyze the proposed NBA-GNN framework. To be specific, we show that (a) our NBA-GNN improves the upper bound for sensitivity-based measures of GNN over-squashing and (b) NBA-GNNs can detect the underlying structure of SBMs even for very sparse graphs.

### 4.1 SENSITIVITY ANALYSIS ON OVER-SQUASHING

We first analyze how NBA-GNNs alleviate the over-squashing issue. To this end, following Topping et al. (2022), we use the Jacobian of node-wise output with respect to another node initial feature as a way to assess over-squashing effect. We first derive the Jacobian quantification of over-squashing for the NBA-GNNs, and then demonstrate that the upper bound for Jacobian is larger than that of the conventional GNNs derived by Topping et al. (2022).

To derive our analysis, we introduce the non-backtracking matrix $B \in \{0, 1\}^{2|\mathcal{E}| \times 2|\mathcal{E}|}$ and the incidence matrix $C \in \mathbb{R}^{2|\mathcal{E}| \times |\mathcal{V}|}$ which describe the NBA-GNN message-passing and node-wise aggregation via linear operation, respectively. To be specific, the backtracking matrix $B$ and the incidence

matrix $C$ are defined as follows:

$$B_{(\ell\to k),(j\to i)} = \begin{cases} 1 & \text{if } k = j, \ell \neq i \\ 0 & \text{otherwise} \end{cases} \quad , \quad C_{(k\to j),i} = \begin{cases} 1 & \text{if } j = i \text{ or } k = i \\ 0 & \text{otherwise} \end{cases} \quad .$$

We also define $D_{out}$ and $D_{in}$ as the out-degree and in-degree matrices of NBA-GNN, respectively, counting the number of outgoing and incoming edges for each edge. These are diagonal matrices with $(D_{out})_{(j\to i),(j\to i)} = \sum_{\ell\to k} B_{(j\to i),(\ell\to k)}$ and $(D_{in})_{(j\to i),(j\to i)} = \sum_{\ell\to k} B_{(\ell\to k),(j\to i)}$. Next, we introduce $\widehat{B}$ as the normalized non-backtracking matrix augmented with self-loops: $\widehat{B} = (D_{out} + I)^{-\frac{1}{2}}(B + I)(D_{in} + I)^{-\frac{1}{2}}$. Finally, we let $\tilde{C}$ denote a matrix where $\tilde{C}_{(k\to j),i} = C_{(k\to j),i} + C_{(j\to k),i}$. Then, one obtains the following sensitivity bound of NBA-GNN.

**Lemma 1.** *Consider two nodes $i, j \in \mathcal{V}$ with distance $T$ given a $(T - 1)$-layer NBA-GNN as described in Equation (3) and Equation (4). Suppose $|\nabla\phi^{(t)}|, |\nabla\sigma| \leq \alpha$, $|\nabla\psi^{(t)}|, |\nabla\rho| \leq \beta$, and $|\nabla\phi| \leq \gamma$ for $0 \leq t < T$. Then the following holds:*

$$\left\| \frac{\partial h_j}{\partial x_i} \right\| \leq (\alpha\beta)^T \gamma (\tilde{C}^\top \widehat{B}^{(T-1)} \tilde{C})_{j,i}.$$

We provide the proof in Appendix C. Lemma 1 states how *the over-squashing effect is controlled by the power of $\widehat{B}$.* Consequently, one can infer that increasing the upper bound likely leads to mitigation of the GNN over-squashing effect (Topping et al., 2022).

From this motivation, we provide an analysis to support our claim on how the NBA-GNN suffers less from over-squashing effect due to its larger sensitivity bound.

**Theorem 1.** *Let $\widehat{A}$ denote the degree-normalized matrix. Then, for any pair of nodes $i, j \in \mathcal{V}$ with distance $T$, the sensitivity bound of NBA-GNN is larger than that of conventional GNNs (Topping et al., 2022), i.e., $(\tilde{C}^\top \widehat{B}^{T-1} \tilde{C})_{j,i} \geq (\widehat{A}^T)_{j,i}$. For $d$-regular graphs, $(\tilde{C}^\top \widehat{B}^{T-1} \tilde{C})_{j,i}$ decays slower by $O(d^{-T})$, while $(\widehat{A}^T)_{j,i}$ decays with $O((d + 1)^{-T})$.*

Note that $(\alpha\beta)^T (\widehat{A}^T)_{j,i}$ provides an upper bound for the sensitivity term in conventional GNNs (Topping et al., 2022). We provide the proof in Appendix C. Our proof is based on comparing the degree-normalized number of non-backtracking and simple walks from node $i$ to node $j$.

## 4.2 EXPRESSIVE POWER OF NBA-GNN IN THE LENS OF SBMs

In the literature on the expressive capabilities of GNNs, comparisons with the well-known $k$-WL test are common. However, even when certain models surpass the $k$-WL test in performance, evaluating their relative merits remains a nontrivial task. Furthermore, due to the substantial performance gap between the 1-WL and 3-WL tests, many algorithms fall into a range between these two tests, making it more difficult to compare them with each other. It is also worth noting that comparing GNNs with the WL test does not always accurately reflect their performance on real-world datasets.

To address these issues, several studies have turned to spectral analysis of GNNs. From a spectral viewpoint, GNNs can be seen as functions of the eigenvectors and eigenvalues of the given graph. NT & Maehara (2019) showed that GNNs operate as low-pass filters on the graph spectrum, and Balcilar et al. (2020) analyzed the use of various GNNs as filters to extract the relevant graph spectrum and measure their expressive power. Moreover, Oono & Suzuki (2020) argue that the expressive power of GNNs is influenced by the topological information contained in the graph spectrum.

The eigenvalues and the corresponding adjacency matrix eigenvectors play a pivotal role in establishing the fundamental limits of community detection in SBM, as evidenced by Abbe et al. (2015), Abbe & Sandon (2015a), Hajek et al. (2016), Yun & Proutiere (2016), and Yun & Proutière (2019). The adjacency matrix exhibits a spectral separation property, and an eigenvector containing information about the assignments of the vertex community becomes apparent (Lei & Rinaldo, 2015). Furthermore, by analyzing the eigenvalues of the adjacency matrix, it is feasible to determine whether a graph originates from the Erdős–Rényi (ER) model or the SBM (Erdős et al., 2013; Avrachenkov et al., 2015). However, these spectral properties are particularly salient when the average degree of the graph satisfies $\Omega(\log n)$. For graphs with average degrees $o(\log n)$, vertices with higher degrees

Table 1: Comparison of conventional MPNNs and GNNs in the long-range graph benchmark, with and without Laplacian positional encoding. We also denote the relative improvement by Imp.

| Model | Peptides-func | | Peptides-struct | | PascalVOC-SP | |
|---|---|---|---|---|---|---|
| | AP $\uparrow$ | Imp. | MAE $\downarrow$ | Imp. | F1 $\uparrow$ | Imp. |
| GCN | $0.5930 \pm {\scriptstyle 0.0023}$ | | $0.3496 \pm {\scriptstyle 0.0013}$ | | $0.1268 \pm {\scriptstyle 0.0060}$ | |
| + NBA | $0.6951 \pm {\scriptstyle 0.0024}$ | +17% | $0.2656 \pm {\scriptstyle 0.0009}$ | +22% | $0.2537 \pm {\scriptstyle 0.0054}$ | +100% |
| + NBA+LapPE | $\mathbf{0.7206} \pm {\scriptstyle \mathbf{0.0028}}$ | +22% | $\mathbf{0.2472} \pm {\scriptstyle \mathbf{0.0008}}$ | +29% | $\mathbf{0.3005} \pm {\scriptstyle \mathbf{0.0010}}$ | +137% |
| GIN | $0.5498 \pm {\scriptstyle 0.0079}$ | | $0.3547 \pm {\scriptstyle 0.0045}$ | | $0.1265 \pm {\scriptstyle 0.0076}$ | |
| + NBA | $0.6961 \pm {\scriptstyle 0.0045}$ | +27% | $0.2534 \pm {\scriptstyle 0.0025}$ | +29% | $0.3040 \pm {\scriptstyle 0.0119}$ | +140% |
| + NBA+LapPE | $\mathbf{0.7071} \pm {\scriptstyle \mathbf{0.0067}}$ | +29% | $\mathbf{0.2424} \pm {\scriptstyle \mathbf{0.0010}}$ | +32% | $\mathbf{0.3223} \pm {\scriptstyle \mathbf{0.0010}}$ | +155% |
| GatedGCN | $0.5864 \pm {\scriptstyle 0.0077}$ | | $0.3420 \pm {\scriptstyle 0.0013}$ | | $0.2873 \pm {\scriptstyle 0.0219}$ | |
| + NBA | $0.6429 \pm {\scriptstyle 0.0062}$ | +10% | $0.2539 \pm {\scriptstyle 0.0011}$ | +26% | $0.3910 \pm {\scriptstyle 0.0010}$ | +36% |
| + NBA+LapPE | $\mathbf{0.6982} \pm {\scriptstyle \mathbf{0.0014}}$ | +19% | $\mathbf{0.2466} \pm {\scriptstyle \mathbf{0.0012}}$ | +28% | $\mathbf{0.3969} \pm {\scriptstyle \mathbf{0.0027}}$ | +38% |

predominate, affecting eigenvalues and complicating the discovery of the underlying structure of the graph (Benaych-Georges et al., 2019).

In contrast, the non-backtracking matrix exhibits several advantageous properties even for constant-degree cases. In (Stephan & Massoulié, 2022), the non-backtracking matrix demonstrates a spectral separation property and establishes the presence of an eigenvector containing information about vertex community assignments, when the average degree only satisfies $\omega(1)$ and $n^{o(1)}$. Furthermore, Bordenave et al. (2015) have demonstrated that by inspecting the eigenvalues of the non-backtracking matrix, it is possible to discern whether a graph originates from the ER model or the SBM, even when the graph's average degree remains constant. This capability sets NBA-GNNs apart and enhances their performance in both node and graph classification tasks, especially in sparse settings. These lines of reasoning lead to the formulation of the following theorem.

**Theorem 2.** *(Informal) Assume that the average degree in the stochastic block model satisfies the conditions of being at least $\omega(1)$ and $n^{o(1)}$. In such a scenario, the NBA-GNN can map from graph $\mathcal{G}$ to node labels.*

**Theorem 3.** *(Informal) Suppose we have a pair of graphs with a constant average degree, one generated from the stochastic block model and the other from the Erdős–Rényi model. In this scenario, the NBA-GNN is capable of distinguishing between them.*

For an in-depth exploration of this argument, please refer to Appendix D. The rationale behind these valuable properties of the non-backtracking matrix $B$ in sparse scenarios lies in the fact that the matrix $B^k$ exhibits similarity to the $k$-hop adjacency matrix, while $A^k$ is mainly influenced by high-degree vertices. For these reasons, NBA-GNNs would outperform traditional GNNs in both node and graph classification tasks, particularly in sparse graph environments.

## 5 EXPERIMENT

In this section, we assess the effectiveness of NBA-GNNs across multiple benchmarks on graph classification, graph regression, and node classification tasks. Our method shows competitive performance compared to graph Transformers within long-range graph benchmarks, and robust improvements in handling transductive node classification tasks. We also conduct ablation studies to verify our method, and provide experimental details in Appendix E.

### 5.1 LONG-RANGE GRAPH BENCHMARK

The long-range graph benchmark (LRGB) (Dwivedi et al., 2022) is a set of tasks that require learning long-range interactions. We validate our method using three datasets from the LRGB benchmark: graph classification (Peptides-func), graph regression (Peptides-struct), and node classification (PascalVOC-SP). We use performance scores from (Dwivedi et al., 2022) and from each baseline papers: subgraph based GNNs (Abu-El-Haija et al., 2019; Michel et al., 2023; Giusti et al.,

Table 2: Evaluation of NBA-GNN on the LRGB benchmark. We color the **first-**, **second-** and **third**-best results. Performance within a standard deviation of one another is considered equal. Non-reported values are denoted by -.

| Method | Model | Peptides-func AP↑ | Peptides-struct MAE↓ | VOC-SP F1↑ |
|---|---|---|---|---|
| GNNs | GCN | $0.5930 \pm 0.0023$ | $0.3496 \pm 0.0013$ | $0.1268 \pm 0.0060$ |
|  | GIN | $0.5498 \pm 0.0079$ | $0.3547 \pm 0.0045$ | $0.1265 \pm 0.0076$ |
|  | GatedGCN | $0.5864 \pm 0.0077$ | $0.3420 \pm 0.0013$ | $0.2873 \pm 0.0219$ |
|  | GatedGCN+PE | $0.6069 \pm 0.0035$ | $0.3357 \pm 0.0006$ | $0.2860 \pm 0.0085$ |
| Subgraph GNNs | MixHop-GCN | $0.6592 \pm 0.0036$ | $0.2921 \pm 0.0023$ | $0.2506 \pm 0.0133$ |
|  | MixHop-GCN+LapPE | $0.6843 \pm 0.0049$ | $0.2614 \pm 0.0023$ | $0.2218 \pm 0.0174$ |
|  | PathNN | $0.6816 \pm 0.0026$ | $0.2545 \pm 0.0032$ | - |
|  | CIN++ | $0.6569 \pm 0.0117$ | $0.2523 \pm 0.0013$ | - |
| Transformers | Transformer+LapPE | $0.6326 \pm 0.0126$ | $0.2529 \pm 0.0016$ | $0.2694 \pm 0.0098$ |
|  | GraphGPS+LapPE | $0.6535 \pm 0.0041$ | $0.2500 \pm 0.0005$ | **$0.3748 \pm 0.0109$** |
|  | SAN+LapPE | $0.6384 \pm 0.0121$ | $0.2683 \pm 0.0043$ | $0.3230 \pm 0.0039$ |
|  | Exphormer | $0.6527 \pm 0.0043$ | $0.2481 \pm 0.0007$ | **$0.3966 \pm 0.0027$** |
|  | Graph MLP-Mixer/ViT | $0.6970 \pm 0.0080$ | **$0.2449 \pm 0.0016$** | - |
| Rewiring methods | DIGL+MPNN | $0.6469 \pm 0.0019$ | $0.3173 \pm 0.0007$ | $0.2824 \pm 0.0039$ |
|  | DIGL+MPNN+LapPE | $0.6830 \pm 0.0026$ | $0.2616 \pm 0.0018$ | $0.2921 \pm 0.0038$ |
|  | DRew-GCN+LapPE | **$0.7150 \pm 0.0044$** | $0.2536 \pm 0.0015$ | $0.1851 \pm 0.0092$ |
|  | DRew-GIN+LapPE | **$0.7126 \pm 0.0045$** | $0.2606 \pm 0.0014$ | $0.2692 \pm 0.0059$ |
|  | DRew-GatedGCN+LapPE | $0.6977 \pm 0.0026$ | $0.2539 \pm 0.0007$ | $0.3314 \pm 0.0024$ |
| **NBA-GNNs (Ours)** | NBA-GCN | $0.6951 \pm 0.0024$ | $0.2656 \pm 0.0009$ | $0.2537 \pm 0.0054$ |
|  | NBA-GCN+LapPE | **$0.7207 \pm 0.0028$** | **$0.2472 \pm 0.0008$** | $0.3005 \pm 0.0010$ |
|  | NBA-GIN | $0.6961 \pm 0.0045$ | $0.2775 \pm 0.0057$ | $0.3040 \pm 0.0119$ |
|  | NBA-GIN+LapPE | **$0.7071 \pm 0.0067$** | **$0.2424 \pm 0.0010$** | $0.3223 \pm 0.0063$ |
|  | NBA-GatedGCN | $0.6429 \pm 0.0062$ | $0.2539 \pm 0.0011$ | **$0.3910 \pm 0.0010$** |
|  | NBA-GatedGCN+LapPE | $0.6982 \pm 0.0014$ | **$0.2466 \pm 0.0012$** | **$0.3969 \pm 0.0027$** |

Table 3: Comparison of conventional GNNs and their NBA-GNN counterpart on transductive node classification tasks, with and without positional encoding. We mark the best numbers in bold.

| Model | Cora | CiteSeer | PubMed | Texas | Wisconsin | Cornell |
|---|---|---|---|---|---|---|
| GCN | $0.8658 \pm 0.0060$ | $0.7532 \pm 0.0134$ | $0.8825 \pm 0.0042$ | $0.6162 \pm 0.0634$ | $0.6059 \pm 0.0438$ | $0.5946 \pm 0.0662$ |
| +NBA | $\mathbf{0.8722} \pm \mathbf{0.0095}$ | $0.7585 \pm 0.0175$ | $0.8826 \pm 0.0044$ | $\mathbf{0.7108} \pm \mathbf{0.0796}$ | $\mathbf{0.7471} \pm \mathbf{0.0386}$ | $0.6108 \pm 0.0614$ |
| +NBA+LapPE | $0.8720 \pm 0.0129$ | $\mathbf{0.7609} \pm \mathbf{0.0186}$ | $\mathbf{0.8827} \pm \mathbf{0.0048}$ | $0.6811 \pm 0.0595$ | $\mathbf{0.7471} \pm \mathbf{0.0466}$ | $\mathbf{0.6378} \pm \mathbf{0.0317}$ |
| GraphSAGE | $0.8632 \pm 0.0158$ | $0.7559 \pm 0.0161$ | $0.8864 \pm 0.0030$ | $0.7108 \pm 0.0556$ | $0.7706 \pm 0.0403$ | $0.6027 \pm 0.0625$ |
| +NBA | $\mathbf{0.8702} \pm \mathbf{0.0083}$ | $0.7586 \pm 0.0213$ | $\mathbf{0.8871} \pm \mathbf{0.0044}$ | $0.7270 \pm 0.0905$ | $\mathbf{0.7765} \pm \mathbf{0.0508}$ | $\mathbf{0.6459} \pm \mathbf{0.0691}$ |
| +NBA+LapPE | $0.8650 \pm 0.0120$ | $\mathbf{0.7621} \pm \mathbf{0.0172}$ | $0.8870 \pm 0.0037$ | $\mathbf{0.7486} \pm \mathbf{0.0612}$ | $0.7647 \pm 0.0531$ | $0.6378 \pm 0.0544$ |
| GAT | $0.8694 \pm 0.0119$ | $0.7463 \pm 0.0159$ | $0.8787 \pm 0.0046$ | $0.6054 \pm 0.0386$ | $0.6000 \pm 0.0491$ | $0.4757 \pm 0.0614$ |
| +NBA | $\mathbf{0.8722} \pm \mathbf{0.0120}$ | $0.7549 \pm 0.0171$ | $\mathbf{0.8829} \pm \mathbf{0.0043}$ | $0.6622 \pm 0.0514$ | $0.7059 \pm 0.0562$ | $\mathbf{0.5838} \pm \mathbf{0.0558}$ |
| +NBA+LapPE | $0.8692 \pm 0.0098$ | $\mathbf{0.7561} \pm \mathbf{0.0175}$ | $0.8822 \pm 0.0047$ | $\mathbf{0.6730} \pm \mathbf{0.0348}$ | $\mathbf{0.7314} \pm \mathbf{0.0531}$ | $0.5784 \pm 0.0640$ |
| GatedGCN | $0.8477 \pm 0.0156$ | $0.7325 \pm 0.0192$ | $\mathbf{0.8671} \pm 0.0060$ | $0.6108 \pm 0.0652$ | $0.5824 \pm 0.0641$ | $0.5216 \pm 0.0987$ |
| +NBA | $\mathbf{0.8523} \pm \mathbf{0.0095}$ | $\mathbf{0.7405} \pm \mathbf{0.0187}$ | $0.8661 \pm 0.0035$ | $0.6162 \pm 0.0490$ | $0.6431 \pm 0.0356$ | $\mathbf{0.5649} \pm \mathbf{0.0532}$ |
| +NBA+LapPE | $0.8517 \pm 0.0130$ | $0.7379 \pm 0.0193$ | $0.8661 \pm 0.0047$ | $\mathbf{0.6243} \pm \mathbf{0.0467}$ | $\mathbf{0.6569} \pm \mathbf{0.0310}$ | $0.5405 \pm 0.0785$ |

2023), graph Transformers (Kreuzer et al., 2021; Rampášek et al., 2022; Shirzad et al., 2023; He et al., 2023), and graph rewiring methods (Gasteiger et al., 2019; Gutteridge et al., 2023). For NBA-GNNs and NBA-GNNs with begrudgingly backtracking, we report the one with better performance. Furthermore, LapPE, i.e., Laplacian positional encoding (Dwivedi et al., 2020), is applied, as it enhances the performance of NBA-GNNs in common cases.

As one can see in Table 1, NBA-GNNs show improvement regardless of the combined backbone GNNs, i.e., GCN (Kipf & Welling, 2016), GIN (Xu et al., 2019), and GatedGCN (Bresson & Laurent, 2017). Furthermore, even against a variety of baselines in Table 2, at least one NBA-GNNs shows competitive performance with the best baseline in LRGB. Also, it is noteworthy that the improvement of NBA-GNNs is higher in dense graphs, `PascalVOC-SP` having an average degree of 8 while `Peptides-func`, `Peptides-struct` has an average degree of 2.

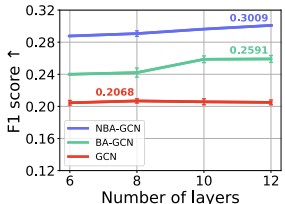

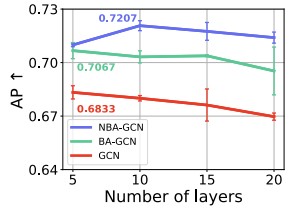

| Model | BG. | AP ↑ | MAE ↓ |
|---|---|---|---|
| GCN | ✗ | 0.7015 | **0.2472** |
| | ✓ | **0.7207** | 0.2547 |
| GIN | ✗ | 0.6825 | **0.2424** |
| | ✓ | **0.7071** | 0.2479 |
| GatedGCN | ✗ | 0.6710 | **0.2466** |
| | ✓ | **0.6982** | 0.2489 |

(a) F1 for changes in number of layers in `PascalVOC-SP`

(b) AP for changes in number of layers in `Peptides-func`

(c) Performance of begrudgingly, non-backtracking in sparse graphs

Figure 5: Ablation studies on the components of NBA-GNN.

## 5.2 TRANSDUCTIVE NODE CLASSIFICATION TASKS

We conduct experiments on three citation networks (Cora, CiteSeer, and Pubmed) (Sen et al., 2008), and three heterophilic datasets (Texas, Wisconsin, and Cornell) (Pei et al., 2019) for transductive node classification. In our experiments, we employed various GNN architectures (Kipf & Welling, 2016; Hamilton et al., 2017; Veličković et al., 2018; Li et al., 2016), along with their corresponding NBA versions, as illustrated in Table 3. The results indicate that the NBA operation improves the performance of all GNN variants. Specifically, it demonstrates significant enhancements, particularly on heterophilic datasets. Given that related nodes in heterophilic graphs are often widely separated (Zheng et al., 2022), the ability of NBA-GNN to alleviate over-squashing plays a vital role in classifying nodes in such scenarios.

## 5.3 ABLATION STUDIES

Here, we conduct ablation studies to empirically verify our framework. For simplicity, we use BA for backtracking GNNs and BG for begrudgingly backtracking GNNs. All experiments are averaged over 3 seeds. We use hyper-parameters for GCN from Tönshoff et al. (2023).

**Non-backtracking vs. backtracking GNNs.** We first verify whether the performance improvements indeed stem from the non-backtracking updates. To this end, we compare our NBA-GNN with a backtracking variant, coined BA-GNN. To be specific, BA-GNN allows backtracking update prohibited in NBA-GNN , i.e., using $h_{i \to j}^{(\ell)}$ to update $h_{j \to i}^{(\ell+1)}$. From Figures 5a and 5b, one can observe how NBA-GNN outperforms the BA-GNN consistently irrelevant to the numbers of layers. Intriguingly, one can also observe that BA-GNN outperforms the naïve backbone, i.e., GCN, consistently.

**Begrudgingly backtracking updates in sparse graphs.** Additionally, we investigate the effect of begrudgingly backtracking in sparse graphs, i.e., `Peptides-func`, and `Peptides-struct`, in Figure 5c. One can see that begrudgingly backtracking is effective in `Peptides-func`, and shows similar performance in `Peptides-struct` (Note that the `PascalVOC-SP` does not have a vertex with degree one).

## 6 CONCLUSION

We have introduced message passing framework applicable to any GNN architectures, bringing non-backtracking into graph neural networks. As theoretically shown, NBA-GNNs mitigate over-squashing in terms of sensitivity, and enhance its expressive power for both node and graph classification tasks on SBMs. Additionally, we demonstrate that NBA-GNNs achieve competitive performance on the LRGB benchmark and outperform conventional GNNs across various tasks.

**Limitations.** The space complexity of our framework is larger than the conventional GNNs. It creates $2|\mathcal{E}|$ messages considering directions, and $\sum_{i \in \mathcal{V}} \binom{d_i}{2}$ connections between them where $d_i$ is the degree of a node $i$. The time complexity is also $\sum_{i \in \mathcal{V}} \binom{d_i}{2}$, since we do not add additional message-passing to the backbone GNN. We provide more detail of the space and time complexity in Appendix F. Therefore, investigation of space complexity reduction for our work would be an interesting future work.

**Ethics Statement.** The authors are committed to upholding the ICLR Code of Ethics and maintaining the highest ethical standards in research and scholarly activities. Additionally, it is crucial to acknowledge the potential ethical considerations associated with the utilization of our model. Although the model can offer valuable insights and advantages, there exists a possibility of misuse or unintended consequences. For instance, it could be exploited to extract sensitive information, such as affiliations or political inclinations, from social networks. Therefore, it is important to apply such models carefully and actively work towards preventing any adverse side effects. Responsible implementation is essential to ensure that the potential benefits of our work are maximized while minimizing any potential risks or adverse implications.

**Reproducibility.** The code for all experiments conducted in this paper is included in the accompanying zip file. We also provide comprehensive proofs for the theoretical analysis in Appendices B, C, and D. Further information about the experiments, including details about the datasets and models, can be found in Section 5. For a deeper dive into experiment specifics such as hyperparameter settings, please refer to Appendix E.

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
