# OpenReview forum: "Non-backtracking Graph Neural Networks"
_ICLR.cc/2024/Conference — Submitted to ICLR 2024_

### Official Review · Reviewer_apAV · 2023-10-31

**Soundness:** 3 good
**Presentation:** 3 good
**Contribution:** 2 fair
**Rating:** 5
**Confidence:** 5

**Summary:**

The paper investigates the message-passing scheme of GNNs and introduces non-backtracking GNNs, which avoid that a message passed from a node $u$ to a node $v$ contributes to the message that is passed from $v$ to $u$ in the next layer. The authors introduce established concepts for random walks and their analysis to GNNs and make formal connections to oversquashing. The experimental evaluation shows clear improvements of the approach over their standard counterpart and SOTA results on various data sets.

**Strengths:**

1. An established and well-investigated concept from other fields is used to improve GNNs.
2. The experimental evaluation is convincing and shows improvements, particularly for long-range task datasets and heterophilic node classification.
3. The paper is well written and illustrated by figures.

**Weaknesses:**

1. The novelty of the introduced techniques is limited. The authors do not sufficiently discuss closely related works:
   - Non-backtracking concepts in walk-based graph learning: The problem of backtracking has been investigated for random walk kernels, see:
      * Pierre Mahé, Nobuhisa Ueda, Tatsuya Akutsu, Jean-Luc Perret, Jean-Philippe Vert: Extensions of marginalized graph kernels. ICML 2004
      * Furqan Aziz, Richard C. Wilson, Edwin R. Hancock: Backtrackless Walks on a Graph. IEEE Trans. Neural Networks Learn. Syst. 24(6): 977-989 (2013)

      The first paper by Mahé et al. proposes a transformation of an undirected input graph to a directed graph, where each undirected edge is represented by two nodes reflecting the two ways of traversing the edge. These nodes are connected such that walks with backtracking are not possible. The construction and the idea are highly similar to the method described in the paper under review.
   - The paper "Zhengdao Chen, Lisha Li, Joan Bruna: Supervised Community Detection with Line Graph Neural Networks. ICLR 2019" (cited but not sufficiently acknowledged) introduces a similar idea of avoiding backtracking to GNNs. A GNN implicitly performing message-passing on a directed line graph is proposed, conceptually highly similar to the technique proposed in the paper under review. Moreover, its strength for learning on graphs generated via the stochastic block model is investigated and explored.
   - The paper "Rongqin Chen, Shenghui Zhang, Leong Hou U, Ye Li: Redundancy-Free Message Passing for Graph Neural Networks. NeurIPS 2022" is closely related to the proposed method but only cited among others on page 4 and not sufficiently acknowledged. The approach uses simple paths (and cycles), allowing no repeated vertices at all instead of no repetition in subpaths of length two as the non-backtracking approach. Moreover, it investigates the link to oversquashing via the same techniques based on the Jacobian. A more detailed discussion of the differences compared to this work is necessary.

2. Analysis of expressive power: The section argues that spectral analysis of GNNs overcomes the issues of Wesifeiler-Leman-based expressivity results. I cannot follow the reasoning. While focusing on spectral analysis allows to draw from existing results on non-backtracking walks, I have difficulties understanding Theorems 2 and 3. What exactly is the learning task in Theorem 2? How to interpret "can accurately map from graph $\mathcal{G}$ to node labels"? What is the influence of the learnable parameters of the GNN? Most importantly, it is unclear whether standard GNNs with WL expressivity cannot solve these learning tasks, at least theoretically.

3. The authors argue that non-backtracking GNNs reduce redundancy. However, it is not discussed to what extent this is possible using the proposed method. While the illustrating examples are trees, in graphs with cycles, redundancy still occurs. A natural generalization would be to avoid backtracking within $k$ hops. A discussion of this could strengthen the paper.

4. The space and time complexity increases compared to standard GNNs.

5. Experimental evaluation:
   - The reported results in the tables are the maximum of two variants, which is slightly unfair. Please report the results separately.

Minor remarks:
  - The caption of Figure 3 contains several repetitions

**Questions:**

1. How does the expressivity of NBA-GNNs (e.g., NBA-GIN) compare to GIN?
2. Can you clarify the relation to other works (see weaknesses 1)?
3. Can the approach be extended to avoid backtracking within $k$ hops?
4. What is the intuition as to why begrudgingly backtracking should work better? Does this mean that redundancy is not inherently problematic?
5. LapPE enhances the performance for the long-range task. Are there results for GIN/GCN with LapPE?

---

> ### Author Response · Authors · 2023-11-17
> **Response to Reviewer apAV (1/2)**
>
> Dear Reviewer $\textcolor{lime}{apAV}$,
>
> We sincerely appreciate your valuable and active feedback for improving our work. We hope we have addressed every question. Please feel free to let us know if there still are concerns not resolved to re-evaluate/re-rate our paper.
>
> ---
>
> **W1, Q2** The authors do not sufficiently discuss closely related works
>
> We appreciate your notification and explanation about closely related works. We have discussed them in the general response 1, related works section, and added the detailed difference in Appendix A, as mentioned in general response 1. We clarify the distinction as follows:
>
> **Mahé et al**.: We note that the construction of the graph in Mahé uses both node and edge representation, while **ours only uses edge-wise representation for non-backtracking**. Moreover, CINs [1, 2] could be considered an extent of Mahe et al., with message-passing between different dimension cell complexes.
>
> **LGNN (Chen et al. `19)**: A notable distinction between NBA-GNN and LGNN is that LGNN employs the non-backtracking operator inside a layer. When layers are stacked, redundancy is not addressed at all, and consequently cannot alleviate over-squashing. However, **NBA-GNN keeps the non-backtracking characteristic across all layers to reduce redundant messages** and theoretically shows that the sensitivity upper bound is bigger.
>
> **RFGNN (Chen et al. `22)**: We agree that the motivation is very similar. However, while RFGNN constructs a tree of non-backtracking paths from all nodes, NBA-GNN **considers non-backtracking by edge adjacency** through all layers. Thanks to this, while RFGNN suffers from a space and time complexity of $\mathcal{O} (\vert V \vert^{k+1})$, **our NBA-GNN only requires $d_{avg} \vert\mathcal{E}\vert$, irrelevant to the number of layers $k$**. This allows our NBA-GNN scalable to larger graph datasets, compared to RFGNN. Furthermore, in terms of theory, our work differs by **stating the sensitivity upper bound of non-backtracking update**, not comparing the relative inference of paths.
>
> [1] Bodnar et al., Weisfeiler and Lehman Go Cellular: CW Networks
>
> [2] Giusti et al., CIN++: Enhancing topological message passing
>
> ---
>
> **W2-1** Analysis of expressive power - The exact learning task in thm2, how to interpret “can accurately map graph $\mathcal{G}$ to node labels"?
>
> The learning task of theorem 2 is the node classification. In this theorem, we are discussing the ability of NBA-GNNs that classify each node in one graph, even in the very sparse case.
>
> ---
>
> **W2-2** Analysis of expressive power - It is unclear whether standard GNNs with WL expressivity cannot solve these learning tasks, at least theoretically
>
> From the WL perspective, NBA-GNN is **as powerful as the WL test** (and other equivalently powerful GNN structures such as GIN) under the assumption that the aggregation and update functions in the layers of NBA-GNN, as specified in equation (3) in our paper, are injective. To illustrate this, consider the scenario where NBA-GNN maps graphs $G$ and $H$ to identical embeddings. In this case, an order of oriented edges $e_i\in E_{G}$, and $d_i\in E_{H}$ exists, satisfying $f(e_i)=f(d_i)$, where $f$ is the non-backtracking message passing update function. This implies that {$j|e_j\in N(e_i)$}={$j|d_j\in N(d_i)$} holds for any given $i$. By the definition of oriented edges, there also exists an order of vertices $u_i\in V_{G}$, and $v_i\in V_{H}$ such that {$j|u_j\in N(u_i)$}={$j|v_j\in N(v_i)$} for all $i$. Consequently, the 1-WL test determines that $G$ and $H$ are isomorphic, establishing that NBA-GNN is equally powerful as the WL test.
>
> However, while we explore the previous works on the expressivity of GNNs, we found that the WL test has some limitations in that it cannot measure the expressive power concerning node classification tasks, and still, it is difficult to compare the proposed algorithms due to the wide performance gap between the 1-WL and 3-WL tests. Therefore, we focus on the spectral analysis in this work to analyze the theoretical performance of NBA-GNNs.
>
> ---
>
> **W3, Q3** Can NBA-GNN be extended to avoid redundancy in cycles by non-backtracking within $k$ hop?
>
> No, NBA-GNN considers redundancy by two consecutive steps in a random walk. Efficiently resolving redundancy occurred by cycles is an interesting future work, since the VOC-SP dataset contains a massive number of cycles.

---

> ### Author Response · Authors · 2023-11-17
> **Response to Reviewer apAV (2/2)**
>
> **W4** Space and time complexity increases compared to standard GNNs.
>
> Yes, the space and time complexity increases compared to the backbone GNN. However, **NBA-GNN achieves superiority over complexity compared to previous work that alleviates over-squashing**. We provide the preprocessing time, test time per epoch, and memory usage below.
>
> Table 1 - Theoretical & actual time complexity for preprocessing($b$: branching factor, $k$: number of layers, and $d_{avg}$: average degree of nodes)
>
> |                             | PathNN-SP             | DRew              | RFGNN                      | NBA-GNN                         |
> |-----------------------------|-----------------------|-------------------|----------------------------|---------------------------------|
> | Theoretical time complexity |$ \|V\| * b^k$           | $\|E\|\sim\|V\|*\|E\|$ | $\|V\|! / \|V-k-1\|!$        | $d_{avg}\|E\|$                          |
> | Actual time                 | 666s                  | 123s              | N/A                        | 68s                             |
> | Content                     | singles shortest path | $k$-hop indices     | $k$-depth non-redundant tree | Non-backtracking edge adjacency |
>
> Table 2 - Average test time per epoch (Tested on a single RTX 3090, the batch size inside the parentheses)
>
> | Model          | peptides-func | peptides-struct | VOC-SP     |
> |----------------|---------------|-----------------|------------|
> | PathNN-SP      | 3.349 (64)    | 2.120 (64)      | N/A        |
> | PathNN-SP+     | 5.156 (64)    | 1.417 (64)      | N/A        |
> | PathNN-AP      | 5.977 (64)    | 1.372 (64)      | N/A        |
> | GraphGPS+LapPE | 0.639 (128)   | 0.926 (128)     | 1.307 (32) |
> | DRew-GCN+LapPE | 0.933 (128)   | 1.053 (128)     | 6.468 (20) |
> | NBA-GCN+LapPE  | 0.541 (200)   | 0.532 (200)     | 5.866 (30) |
>
> Table 3: Memory usage (Single RTX 3090, the batch size inside the parentheses)
>
> | Model          | peptides-func | peptides-struct | VOC-SP      |
> |----------------|---------------|-----------------|-------------|
> | GCN            | 8.5% (128)    | 13.6% (128)     | 20.12% (32) |
> | PathNN-SP      | OOM (128)     | OOM (128)       | N/A         |
> | PathNN-SP+     | OOM (128)     | OOM (128)       | N/A         |
> | PathNN-AP      | OOM (128)     | OOM (128)       | N/A         |
> | GraphGPS+LapPE | 89.7% (128)   | 88.08% (128)    | 45.62% (32) |
> | DRew-GCN+LapPE | 46.82% (128)  | 46.82% (128)    | OOM (32)    |
> | NBA-GCN+LapPE  | 18.86% (128)  | 16.98% (128)    | 47.30% (32) |
>
> ---
>
> **W5** Experimental evaluation: Table 1 contains the maximum of two variants (NBA, begrudgingly) which seems unfair
>
> Yes, the NBA results presented in Table 1 represent the maximum of non-backtracking and begrudgingly walks. While we have reported this in accordance with the prior work DRew [1], it is worth noting that this approach may be perceived as unfair. We have provided **an ablation study in Figure 5(c) for a comparison between two variants** on the Peptides-func dataset. In response to the reviewer's request, we have also reported the variants in the peptides-struct dataset (given the absence of dangling nodes in the VOC-SP dataset). Despite these considerations, our results still demonstrate improvements compared to the baselines.
>
> [1] Gutteridge et al., DRew: Dynamically Rewired Message Passing with Delay, ICML 2023
>
> ---
>
> **Q1** How does the expressivity of NBA-GNNs (e.g., NBA-GIN) compare to GIN?
>
> Building on a similar argument as discussed in the response to **W2-2**, if we assume that the aggregation and update functions in the layers of NBA-GNN, as specified in equation (3) in our paper, are injective, we can infer that NBA-GNN is at least as powerful as GIN or NBA-GIN.
>
> ---
>
> **Q4** intuition of begrudgingly backtracking should work better and redundancy
>
> The underlying idea is to address the information loss caused by the excessive reduction of redundancy, particularly in proximity to dangling nodes—those with only one neighbor. When a message reaches a dangling node, it becomes unable to propagate information to other nodes, leading to information loss. To mitigate this, we employ begrudgingly backtracking.
>
> ---
>
> **Q5** Results for GIN/GCN with LapPE
>
> Yes, there are. As one can see in Table 1 in the paper, we have provided the results for GCN, GIN with and without LapPE, and LapPE. As the reviewer assumed, the LapPE indeed enhances the performance for long-range tasks.
> M1 The caption of Figure 3 contains several repetitions
> Thank you for your careful review. We have modified Figure 3 and its caption to make it clear.
>
> ---
>
> **M1** The caption of Figure 3 contains several repetitions
>
> Thank you for your careful review. We have modified Figure 3 and its caption to make it clear.

---

> > ### Comment · Reviewer_apAV · 2023-11-22
> >
> > I appreciate the detailed reply and additional results provided by the authors.
> >
> > **W1/Q2** I still think that the novelty is limited. First, the claimed difference compared to the preprocessing proposed by Mahé et al. is minor. Note that the node-wise aggregation proposed by the authors is very similar to what would happen at the node representations used by Mahé et al. Second, the theoretical contribution / sensitivity analysis remains incremental.
> >
> > **W2-1/2** Does this hold for arbitrary node labels? I think it is necessary to work out the advantage over MPNNs in these tasks more clearly.

---

> > > ### Author Response · Authors · 2023-11-23
> > > **Response to Reviewer apAV**
> > >
> > > We appreciate your thoughtful review and questions again. If there are any remaining concerns, please feel free to bring them to our attention for further evaluation or reevaluation of our paper.
> > >
> > > ---
> > >
> > > **(W1/Q2)-1** The claimed difference compared to the preprocessing proposed by Mahé et al. is minor. Note that the node-wise aggregation proposed by the authors is very similar to what would happen at the node representations used by Mahé et al.
> > >
> > > As we mentioned in global response 1, LGNN was the first to use non-backtracking and we have updated the manuscript accordingly. Although NBA-GNN and Mahé et al share the common idea of non-backtracking, it differs in many details.
> > >
> > > i) **Node set of transformed graph**: Figure 2 of Mahé et al. states that the transformed graph is defined as **$G=(V’, E’)$ where $V’=V\cup E$**, from an input graph $G=(V, E)$. However, our NBA-GNN computation graph can be written as **$G’=(V’, E’)$ where $V’=E$**.
> > >
> > > ii) **Initial representation**: The random walk in Mahé et al must start from a node in the original graph, i.e., V, while the update process of **NBA-GNN only considers edge-wise representations**. Moreover, the edge $(i, j)$ only receives information from node $i$ in Mahé et al, while the edge representation of NBA-GNN is built by using both node $i, j$.
> > >
> > > iii) **Node-wise aggregation**: NBA-GNN **aggregates the set of incoming and outgoing edges for a node representation**, for example, node-wise prediction in the VOC-SP dataset. In Mahé et al, I may have missed the content of node-wise aggregation. But based on IV) of Figure 2, it seems to **only consider random walks starting from a node**, not random walks ending at a node. We would be happy to provide more details if the reviewer could point out the section node-wise aggregation is described.
> > >
> > > ---
> > >
> > > **(W1/Q2)-2** The theoretical contribution/sensitivity analysis remains incremental.
> > >
> > > As we highlighted in global response 2, our contribution lies in being the first **to compare the degree of over-squashing between GNNs, aligning with different random walks**. While previous works have used metrics such as curvature$^{[1]}$ or the effective resistance$^{[2]}$ to assess the graph topology in the sensitivity upper bound, we have interpreted it using random walks. Furthermore, demonstrating the tightness of sensitivity bounds requires unrealistic strong assumptions, e.g., there always exist parameters that set the sensitivity term to zero.
> > >
> > > [1] Topping et al., Understanding over-squashing and bottlenecks on graphs via curvature, ICLR 2022
> > >
> > > [2] Black et al, Understanding over-squashing in GNNs through the Lens of Effective Resistance, ICML 2023
> > >
> > > ---
> > >
> > > **(W2-1/2)-1** Does this hold for arbitrary node labels?
> > >
> > > Yes, the aforementioned argument remains valid for arbitrary node labels, given that the function from end-node features to edge features is injective. This assertion holds true for our model as we define edge features as the concatenation of end-node features. (For the definition of a message initialization function, please refer to Appendix E.1 in the supplementary material.)
> > >
> > > ---
> > >
> > > **(W2-1/2)-2** I think it is necessary to work out the advantage over MPNNs in these tasks more clearly.
> > >
> > > When comparing conventional MPNNs with NBA-GNNs in the context of the WL test, we can affirm their equivalence to the power of the WL test, but determining superiority is challenging under the assumption of injective aggregation and update functions. This challenge arises from the notable performance gap between the 1-WL and 3-WL tests, where many algorithms fall within this range.
> > >
> > > Therefore, instead of comparison with the WL test, we analyze their performance through spectral analysis to demonstrate the superiority of NBA-GNNs. From a spectral perspective, **GNNs function as spectral filters** $^{[3]}$. However, in a very sparse regime, the spectrum of conventional MPNNs—specifically, the eigenvalues and eigenvectors of the adjacency matrix—lacks a spectral separation property, resulting in **the loss of hidden structure information**. In contrast, NBA-GNNs, with the non-backtracking matrix retaining a spectral separation property, **allow for the recovery of hidden structures**.
> > >
> > > [3] Balcilar, Muhammet, et al. "Analyzing the expressive power of graph neural networks in a spectral perspective." Proceedings of the International Conference on Learning Representations (ICLR). 2021.

---

### Official Review · Reviewer_ayH6 · 2023-10-31

**Soundness:** 3 good
**Presentation:** 4 excellent
**Contribution:** 1 poor
**Rating:** 3
**Confidence:** 4

**Summary:**

The article proposes an architecture of graph neural network that is based on the non-backtracking operator on graphs. The authors explain what non-backtracking walks are and define a GNN based on them. They give some theoretical insights proving this model is less sensitive to mixing far features (over-squashing) than ordinary GNNs and that it is expressive enough to be able to classify sparse binary SBMs. They provide a few experiments on benchmarks showing their network does better or as good as benchmarks.

**Strengths:**

The results in table 2 are promising and using non-backtracking updates for GNN seems relevant. This is also supported by theory, in particular on SBM.

The article is well-written.

**Weaknesses:**

I think the contribution this article brings is too small.

Mainly it seems the authors do not know about the work « Supervised Community Detection with LGNN » Chen et al. ICLR19 arxiv:1705.08415. In this article a GNN based on the non-backtracking operator is proposed ; it has features on the edges that are aggregated via the non-backtracking matrix B ; and, if I am right, it is very similar to the NBA-GNN the authors propose. The differences are that it is formulated directly as a GCN and not a generic permutation-invariant GNN ; and it seems more expressive since it also has features on the nodes (which, for instance, permits not to use the trick of begrudgingly updates) and it applies powers of B to aggregate the features (which, as they show, increases the performance).

The theoretical analysis the authors propose is quite light in the sense that theorems 2 and 3 come straightforwardly from Bordenave 15 and Stephan and Massoulié 22. In the sensibility analysis (theorem 1) the improvement given by non-backtracking is quite modest ; considering the spectral properties of B on a model seems better than analyzing the sensibility.

My point of view is that the novelty of this article is restricted to the experiments it proposes and its broader theroretical frame, that was less developped at the time of Chen 19.

**Questions:**

About NBA-GNN on SBM : the theoretical results are only about its expressiveness ; what about the training ? do the authors actually observe that a trained NBA-GNN can correctly classify the nodes ? They could compare to the conjectured optimal performances given by BP on sparse SBM. I guess they would obtain the same as Chen 19.

It would have been interesting to consider NBA-GNN on the CSBM since this model has features.

Another reference the authors may not know : arxiv:1306.5550, that first used the non-backtracking matrix for node classification.

---

> ### Author Response · Authors · 2023-11-17
> **Response to Reviewer ayH6**
>
> Dear Reviewer $\textcolor{green}{ayH6}$,
>
> We sincerely appreciate your valuable and active feedback for improving our work. We hope we have addressed every question. Please feel free to let us know if there still are concerns not resolved to re-evaluate/re-rate our paper.
>
> ---
>
> **W1** Mainly it seems the authors do not know about the work LGNN, and the contribution of the article is too small.
>
> We appreciate your notification and kind explanation about LGNN. Indeed, this is a crucial prior work using the non-backtracking operator and we have added details of it as mentioned in general response 1. However, we highlight that **LGNN does not make any connection between the non-backtracking update and the message redundancy problem**, obviously not alleviating over-squashing. Moreover, because of its k-hop aggregation process within a single layer and sequential updates between the original graph and line graph, LGNN suffers significant computational complexity.
>
> Our NBA-GNN, on the other hand, **resolves redundant messages specifically using the non-backtracking operator**. The experiment results in general response 1 highlight the poor performance and high complexity of LGNN in real-world datasets. Therefore, in comparison, our main contribution is **bridging the non-backtracking operator to message redundancy problems**, with superior space and time scalability compared to previous works alleviating over-squashing.
>
> ---
>
> **W2** The theoretical analysis the authors propose is quite light
>
> As discussed in the general responses 2 and 3, even though our theoretical analysis may exhibit limited novelty, we want to emphasize our primary contributions on the theoretical side: firstly, **the first analysis of specifically connecting the non-backtracking operator with the sensitivity upper bound**, and secondly, the establishment of a connection between prior works on the spectrum non-backtracking matrix and GNNs.
>
> ---
>
> **Q1** They could compare to the conjectured optimal performances given by BP on sparse SBM
>
> Following your advice, we conducted a comparative analysis involving NBA-GNN, BP, and LGNN on two sparse SBMs with distinct parameters: (a) a Binary Assortative SBM (n = 400, C=2, p = 20/n, q = 10/n), and (b) a 5-community Dissociative SBM (n = 400, C=5, p = 0, q = 18/n), where n represents the number of vertices, C is the number of classes, and p and q denote edge probabilities within and across communities, respectively. The table presented below illustrates the average test accuracy across 100 graphs. Notably, BP, known for achieving the information-theoretic threshold, exhibited the best performance, consistent with expectations. Additionally, our NBA-GNN outperformed LGNN in both scenarios.
>
> Table 1 - Average test accuracy on two sparse SBMs
>
> |     |  LGNN  | NBA-GNN |   BP   |
> |-----|:------:|:-------:|:------:|
> | (a) | 0.4885 |  0.4900 | 0.5303 |
> | (b) | 0.1821 |  0.1888 | 0.2869 |
>
> ---
>
> **Q2** It would have been interesting to consider NBA-GNN on the CSBM since this model has features
>
> Yes, we agree that it would be an interesting extension to our works, as we can explore the impact of node features when analyzing the algorithm's performance, under the conditions inferred in real-world datasets
>
> ---
>
> **Q3** reference for first used non-backtracking matrix for node classification: arxiv:1306.5550
>
> We thank you for bringing the overlooked work to our attention. We will definitely incorporate it into the related works section.

---

### Official Review · Reviewer_Q7oz · 2023-10-31

**Soundness:** 3 good
**Presentation:** 4 excellent
**Contribution:** 3 good
**Rating:** 5
**Confidence:** 4

**Summary:**

In the submitted manuscript, the authors notice that representations learned via standard message passing schemes in graph neural networks (GNNs) are dependent on all walks present in graphs. They propose to remove redundancy from the message passing process by considering non-backtracking (and begrudgingly non-backtracking) walks only. This leads them to propose the NBA-GNN, which learns two embeddings per edge, and analyse the potential impact of the over-squashing phenomenon on NBA-GNNs and perform an expressivity analysis. The NBA-GNNs are found to outperform state-of-the-art baselines on several real-world datasets.

**Strengths:**

- The paper is clear and well-written.
- The considered idea is interesting and some theoretical understanding of it is offered by the theoretical results in Section 4.
- The performance in practice of your NBA-GNNs is impressive and compared against a set of relevant baseline models.

**Weaknesses:**

- Comparison to seemingly closely related previous work appears to be lacking (see Question 1).
- NBA-GNNs are prohibitively expensive and the additional expense in terms of computation time is insufficiently explored (see Question 2).
- The theoretical result in Theorem 1 is only a weak indication of alleviated over-smoothing (see Question 3).

**Questions:**

1) There appears to be previous work proposing the use of non-backtracking operators in GNNs, also investigating their model in the context of stochastic blockmodels [1]. I believe it to be pivotal for you to firstly, discuss the differences between their proposed Line Graph Neural Networks and your NBA-GNNs and to secondly, include their LGNN in your experimental baselines to demonstrate whether/which empirical differences exist.

2) The NBA-GNNs you propose are rather expensive in the sense that you learn two embeddings per edge. While it is very good, that you have included a discussion of the additional memory cost in the "Limitations" paragraph, I also believe a discussion of the time complexity of your method to be necessary. Ideally you should evaluate both the time complexity in theory and also provide experimental evaluation of the computation time of your NBA-GNNs compared to the baseline methods.

3) Your result in Theorem 1 appears to be of limited importance to me. The fact that your upper bound on NBA-GNNs is larger than the bound on standard GNNs could either mean that one of the bounded quantities is indeed larger than the other as you suggest, but it could equally well be the case that one of the two bounds is looser than the other in practice, i.e., given the bound on standard GNNs any larger upper bound is trivially also true and would surpass your larger bound, which might put the conclusions you drew from the magnitude of the two upper bounds in jeopardy. It would significantly strengthen your result if you could observe (even if just experimentally) that the considered derivatives are indeed larger for the non-backtracking operator than for the normalised adjacency matrix.


[1] Chen, Zhengdao, Xiang Li, and Joan Bruna. "Supervised community detection with line graph neural networks." ICLR. (2019).

---

> ### Author Response · Authors · 2023-11-17
> **Response to Reviewer Q7oz**
>
> Dear Reviewer $\textcolor{blue}{Q7oz}$,
>
> We sincerely appreciate your valuable and active feedback for improving our work. We hope we have addressed every question. Please feel free to let us know if there still are concerns not resolved to re-evaluate/re-rate our paper.
>
> ---
>
> **W1/Q1** Comparison to seemingly closely related previous work appears to be lacking
>
> We appreciate your notification about the overlooked work and incorporated it in our general response 1, section 2, and appendix A. While LGNN is indeed a significant prior work first using the non-backtracking operator in the graph, it is essential to emphasize that it **does not address the over-squashing issue**.
>
> In contrast, our **NBA-GNN efficiently resolves redundant messages through non-backtracking, as theoretically demonstrated**. The experimental results outlined in the general response 1 underscore the suboptimal performance and high complexity of LGNN in real-world datasets. Therefore, in comparison, our primary contribution lies in **bridging the non-backtracking operator with the over-squashing problem, additionally offering superior space and time scalability compared to previous works alleviating over-squashing**.
>
> ---
>
> **W2/Q2** NBA-GNNs are prohibitively expensive and the additional expense in terms of computation time is insufficiently explored
>
> Yes, undoubtedly there is an increase in time complexity compared to GNNs. However, we emphasize our method is superior to other previous works alleviating over-squashing, on average test time per epoch and preprocessing times. From the two tables below, one can easily conclude that our method does not face bottlenecks in time compared to other baselines, **irrelevant to the number of layers**.
>
> Table 1 - Theoretical & actual time complexity for preprocessing($b$: branching factor, $k$: number of layers, and $d_{avg}$: average degree of nodes)
>
> |                             | PathNN-SP             | DRew              | RFGNN                      | NBA-GNN                         |
> |-----------------------------|-----------------------|-------------------|----------------------------|---------------------------------|
> | Theoretical time complexity |$ \|V\| * b^k$           | $\|E\|\sim\|V\|*\|E\|$ | $\|V\|! / \|V-k-1\|!$        | $d_{avg}\|E\|$                          |
> | Actual time                 | 666s                  | 123s              | N/A                        | 68s                             |
> | Content                     | singles shortest path | $k$-hop indices     | $k$-depth non-redundant tree | Non-backtracking edge adjacency |
>
> Table 2 - Average test time per epoch (Tested on a single RTX 3090, the batch size inside the parentheses)
>
> | Model          | peptides-func | peptides-struct | VOC-SP     |
> |----------------|---------------|-----------------|------------|
> | PathNN-SP      | 3.349 (64)    | 2.120 (64)      | N/A        |
> | PathNN-SP+     | 5.156 (64)    | 1.417 (64)      | N/A        |
> | PathNN-AP      | 5.977 (64)    | 1.372 (64)      | N/A        |
> | GraphGPS+LapPE | 0.639 (128)   | 0.926 (128)     | 1.307 (32) |
> | DRew-GCN+LapPE | 0.933 (128)   | 1.053 (128)     | 6.468 (20) |
> | NBA-GCN+LapPE  | 0.541 (200)   | 0.532 (200)     | 5.866 (30) |
>
>
> ---
>
> **W3/Q3** The theoretical result in Theorem 1 is only a weak indication of alleviated over-squashing
>
> We acknowledge that our upper bound may not be sufficiently tight as we discussed in the general response 2. However, our theorem is **the first to compare the quantity of over-squashing in GNNs, aligning with different random walks**.

---

> > ### Comment · Reviewer_Q7oz · 2023-11-22
> >
> > I want to thank the authors for their detailed response. My concerns about your theoretical results remain. While I agree with the answer you provide, of you likely being the first to prove a result of this kind, I am still not convinced that the result is strong enough to build a sound logical argument on it.  I therefore, choose to maintain my score.

---

> ### Author Response · Authors · 2023-11-23
> **Response to Reviewer Q7oz**
>
> We understand your reservations about the strength of our theoretical results despite our assertion of being the first to compare the degree of over-squashing between GNNs using different random walks. We would like to provide additional insights into the unique contributions of our work:
>
> **1. Novelty in Comparison Methodology**: Our primary contribution lies in introducing a novel approach to compare the degree of over-squashing among GNNs. By aligning with different random walks, we believe we provide a fresh perspective that adds to the existing body of knowledge.
>
> **2. Interpretation through Random Walks**: While previous works have utilized metrics like curvature or effective resistance to assess graph topology in the sensitivity upper bound, we've taken a distinctive path by interpreting it through the lens of random walks. This allows for a unique and complementary understanding of the graph structure.
>
> **3. Realism in Sensitivity Bounds**: We acknowledge your point about the challenges of demonstrating the tightness of sensitivity bounds. Our intention is not to set unrealistic assumptions, but rather to provide a theoretical framework that is more aligned with real-world scenarios, where the sensitivity term may not be consistently set to zero.
>
> We would greatly appreciate any specific feedback or suggestions you may have regarding aspects of the theoretical results that could be strengthened or clarified. Your insights are crucial to us, and we are dedicated to addressing any concerns to the best of our ability.
>
> Once again, thank you for your thoughtful evaluation, and we look forward to the possibility of further discussion to enhance the clarity and robustness of our theoretical contributions.

---

### Official Review · Reviewer_WPCH · 2023-11-01

**Soundness:** 3 good
**Presentation:** 3 good
**Contribution:** 3 good
**Rating:** 6
**Confidence:** 3

**Summary:**

This paper proposes the Non-Backtracking Graph Neural Network (NBA-GNN) to address the redundancy issue in message-passing updates of conventional  GNNs.  NBA-GNN updates messages without incorporating the message from the previously visited node. They also provided a theoretical analysis of the over-squashing phenomenon in the setting of NBA-GNN. The proposed NBA-GNN is empirically evaluated on long-range graph benchmarks and transductive node classification problems, demonstrating competitive performance.

**Strengths:**

1) The proposed NBA-GNN addresses an important issue in GNNs related to the redundancy of message flows and its impact on downstream tasks. Using non-backtracking updates to reduce redundancy is a novel and well-motivated approach.

2) The paper provides a thorough analysis of the redundancy issue, linking it to the over-squashing phenomenon in GNNs.

3) The empirical evaluation of NBA-GNN on long-range graph benchmarks and transductive node classification problems demonstrates its effectiveness and competitive performance compared to conventional GNNs.

**Weaknesses:**

1) The paper lacks a detailed description of the construction of the non-backtracking operator/walk/update and the related implementation in NBA-GNN.

2) The time complexity of processing the non-backtracking seems high, and the preprocessing time is not reported. Additionally, the run time and memory usage of NBA-GNN compared with other GNNs is not reported, making it difficult to evaluate the proposed method comphensively.

**Questions:**

What is the performance of NBA-GNN on the other two datasets in LRGB?

---

> ### Author Response · Authors · 2023-11-17
> **Response to Reviewer WPCH**
>
> Dear Reviewer $\textcolor{red}{WPCH}$,
>
> We sincerely appreciate your valuable and active feedback for improving our work. We hope we have addressed every question. Please feel free to let us know if there still are concerns not resolved to re-evaluate/re-rate our paper.
>
> ---
>
> **W1** The paper lacks a detailed description of the non-backtracking operator
>
> We discussed the non-backtracking operator in the sensitivity analysis, section 4.1. However, as the reviewer suggested, providing a detailed description of using the non-backtracking operator in NBA-GNNs can enhance comprehension. To address this, we added a detailed description in Appendix E regarding the application of the non-backtracking operator in NBA-GNNs.
>
> ---
>
> **W2** Preprocessing time, run time, and memory usage
>
> We thank reviewer $\textcolor{red}{WPCH}$ for suggesting a fair comparison with baselines, considering the complexity of our method. Our method achieves superior results in all three terms, preprocessing time, run time, and memory usage compared to previous works suggested to alleviate over-squashing.
>
> - **Preprocessing time**
>
> We compared the preprocessing time of the peptides-functional dataset. Our preprocessing time is approximately half of DRew, and 1/10 of PathNN-SP as shown in the table below. It is also worth noting that the preprocessing time of previous methods is proportional to the number of layers, **NBA-GNN is irrelevant to the number of layers**.
>
> Table 1 - Theoretical & actual time complexity for preprocessing($b$: branching factor, $k$: number of layers, and $d_{avg}$: average degree of nodes)
> |                             | PathNN-SP             | DRew              | RFGNN                      | NBA-GNN                         |
> |-----------------------------|-----------------------|-------------------|----------------------------|---------------------------------|
> | Theoretical time complexity |$ \|V\| * b^k$           | $\|E\|\sim\|V\|*\|E\|$ | $\|V\|! / \|V-k-1\|!$        | $d_{avg}\|E\|$                          |
> | Actual time                 | 666s                  | 123s              | N/A                        | 68s                             |
> | Content                     | singles shortest path | $k$-hop indices     | $k$-depth non-redundant tree | Non-backtracking edge adjacency |
>
> - **Running time, memory usage**
>
> The average test time per epoch and memory usage have been measured, by reducing the batch size of previous methods except GCN since the paper’s parameters did not fit on a single RTX 3090. We use 8 layers, a hidden dimension of 120 for all experiments, and have written the batch size inside the parentheses.
>
> Table 2 - Average test time per epoch (Tested on a single RTX 3090)
>
> | Model          | peptides-func | peptides-struct | VOC-SP     |
> |----------------|---------------|-----------------|------------|
> | PathNN-SP      | 3.349 (64)    | 2.120 (64)      | N/A        |
> | PathNN-SP+     | 5.156 (64)    | 1.417 (64)      | N/A        |
> | PathNN-AP      | 5.977 (64)    | 1.372 (64)      | N/A        |
> | GraphGPS+LapPE | 0.639 (128)   | 0.926 (128)     | 1.307 (32) |
> | DRew-GCN+LapPE | 0.933 (128)   | 1.053 (128)     | 6.468 (20) |
> | NBA-GCN+LapPE  | 0.541 (200)   | 0.532 (200)     | 5.866 (30) |
>
> Table 3 - Memory usage (Single RTX 3090)
>
> | Model          | peptides-func | peptides-struct | VOC-SP      |
> |----------------|---------------|-----------------|-------------|
> | GCN            | 8.5% (128)    | 13.6% (128)     | 20.12% (32) |
> | PathNN-SP      | OOM (128)     | OOM (128)       | N/A         |
> | PathNN-SP+     | OOM (128)     | OOM (128)       | N/A         |
> | PathNN-AP      | OOM (128)     | OOM (128)       | N/A         |
> | GraphGPS+LapPE | 89.7% (128)   | 88.08% (128)    | 45.62% (32) |
> | DRew-GCN+LapPE | 46.82% (128)  | 46.82% (128)    | OOM (32)    |
> | NBA-GCN+LapPE  | 18.86% (128)  | 16.98% (128)    | 47.30% (32) |
>
> ---
>
> **Q1** What is the performance of NBA-GNN on the other two datasets in LRGB?
>
> We attempted additional experiments for PCQM and Pascal-SP. However, we note that these datasets are quite large compared to peptides and VOC-SP, with approximately 8 to 10 times more data. Regrettably, we only present the experiment results of NBA-GCN on PCQM, one can still see that the performance does not vary much even without tuning.
>
> Table 4 - PCQM PCQM results without tuning
> | PCQM  | Test Hits@1 | Test Hits@3 | Test Hits@10 | Test MRR |
> |-------|:-----------:|:-----------:|:------------:|:--------:|
> | GCN   |    0.1321   |    0.3791   |    0.8256    |  0.3234  |
> | + NBA |    0.1282   |    0.3589   |    0.8186    |  0.3139  |
> | GIN   |    0.1337   |    0.3642   |    0.8147    |  0.3180  |
> | + NBA |    0.1349   |    0.3754   |    0.8347    |  0.3243  |

---

> > ### Comment · Reviewer_WPCH · 2023-11-20
> >
> > Thank you for your comprehensive response. I have no further questions.

---

### Official Review · Reviewer_w5ts · 2023-11-01

**Soundness:** 2 fair
**Presentation:** 3 good
**Contribution:** 2 fair
**Rating:** 3
**Confidence:** 4

**Summary:**

This paper introduces non-backtracking GNNs, which only send messages through non-backtracking paths. A theoretical analysis of their sensitivity and of their expressive power in comparison with conventional GNNs is conducted. Numerical experiments demonstrate the superiority of non-backtracking GNNs in a number of graph machine learning tasks.

**Strengths:**

- The experiments are sufficiently convincing of the superiority of NBA-GNNs in the considered tasks.
- The paper is generally clear and well-written.

**Weaknesses:**

- The paper misses important related work, specifically:
Zhengdao Chen, Lisha Li, Joan Bruna. Supervised Community Detection with Line Graph Neural Networks. ICLR 2019.
This paper was the first to propose the use of the non-backtracking operator in GNNs.
- In light of the above, the proposed architecture is somewhat incremental.
- The theoretical results are not convincing.
    * The claim that NBA-GNNs might help with oversquashing is supported by the assumption, backed only by empirical evidence, that NBA-GNNs have shorter access time than BA-GNNs. A proposition is provided stating that non-backtracking random walks have shorter access times, but it only holds for trees. The conditions of this proposition are too far away from the setup of NBA-GNNs to make a conving claim.
     * As the authors themselves note, Lemma 1 and Theorem 1 only provide upper bounds on the sensitivity of conventional and NBA-GNNs.  It is a stretch to conclude that, because the upper bound for conventional GNNs is lower than the upper bound for NBA-GNNs, the sensitivities behave similarly. I understand that tighter results/lower bounds may not be possible, but considering that this is the main contribution of this paper, it falls somewhat short.
     * The authors do not comment on the fact that for moderate-to-large degree d, the decay rates of the sensitivity upper bounds for conventional and NBA-GNNs will become very close. Moreover, this finding is not in agreement with the empirical findings from Table 1, which show that NBA-GNNs lead to significant performance improvements on dense graphs. This could be regarded as evidence that the sensitivity upper bounds are not very tight.
     * The second theoretical analysis, from Section 4.2, is not very novel, as it is essentially a restatement of theoretical results from the spectral clustering community.

Minor:

- Important references on the expressivity of GNNs from spectral considerations are missing. See e.g. Kanatsoulis et al., and the work of Ribeiro, A.
- The paper can be quite wordy, and repeat many of the same observations.
- While the limitations are briefly discussed, they perhaps deserve a more extensive treatment as the increase in computational complexity is quite high.

**Questions:**

N/A

---

> ### Author Response · Authors · 2023-11-17
> **Response to Reviewer w5ts (1/2)**
>
> Dear Reviewer $\textcolor{violet}{w5ts}$,
>
> We sincerely appreciate your valuable and active feedback for improving our work. We hope we have addressed every question. Please feel free to let us know if there still are concerns not resolved to re-evaluate/re-rate our paper.
>
> ---
>
> **W1/W2**. The paper misses important related works (Chen et al., ICLR 2019)
>
> Thank you for pointing out a significant related work. To incorporate your comment, we conceptually and empirically compare our work and the suggested work (in our General Response 1, Sections 2, and Appendix A). In comparison to the reference (Chen et al., ICLR 2019), our contribution is **the consideration of the non-backtracking operator specifically for the message redundancy problem**, rather than proposing a novel/first non-backtracking architecture in GNNs.
>
> ---
>
> **W3-1**. Proposition 1 only works for trees, being too far away from the setup of NBA-GNNs
>
> Yes, proposition 1 only holds for trees. However, we note that this proposition is only a motivating example to use non-backtracking, not a theoretical claim that NBA-GNN can mitigate over-squashing. We point out additional references [1,2] that investigate faster access time of non-backtracking walks compared to conventional random walks.
>
> [1] Yuan Lin and Zhongzhi Zhang. Non-backtracking centrality based random walk on networks. The Computer Journal, 62(1):63–80, 2019.
>
> [2] Dario Fasino, Arianna Tonetto, and Francesco Tudisco. Hitting times for second-order random walks. European Journal of Applied Mathematics, 34(4):642–666, 2023.
>
> ---
>
> **W3-2**. Lemma 1 and Theorem 1 only provide upper bounds on the sensitivity.
>
> Yes, theorem 1 only provides the upper bound for the sensitivity analysis. However, as mentioned in the general response 2, we would like to emphasize that **the contribution of Theorem 1 is being the first to compare the degree of over-squashing between GNNs, aligning with different random walks**.
>
> ---
>
> **W3-3**. For moderate-to-large degree d, the decay rates of the two sensitivity upper bounds will become very close. Moreover, this finding is not in agreement with the empirical findings from Table 1.
>
> Thank you for the insightful comment. Indeed, our analysis of the regular graphs is more meaningful for graphs with a small degree. Nonetheless, we believe this result to be meaningful since many real-world graphs, e.g., community and molecular graphs, are sparse. Also, we believe that **one cannot compare the performance improvements in different datasets in an apple-to-apple way**, e.g., VOC-SP performance may be more susceptible to over-squashing in GNNs compared to the peptides-func dataset.
>
> ---
>
> **W3-4**. The theoretical analysis in Section 4.2 is not very novel
>
> While one might perceive it that way, our analysis is novel since **it integrates earlier works on the non-backtracking matrix into GNNs**. This integration has the potential to pave the way for advancements in the field, suggesting new perspectives that could lead to innovative insights and methodologies within the GNN framework.
>
> ---
>
> **M1**. References on the expressivity of GNNs from spectral considerations
>
> Thank you once again for bringing the overlooked work to our attention. We will certainly incorporate it into the related works section of our paper. The study by Kanatsoulis and Ribeiro introduces a novel approach to analyzing the expressivity of GNNs, utilizing linear algebraic tools and eigenvalue decomposition of graph operators. Their work showcases that GNNs have the capability to generate distinctive outputs for graphs with varying eigenvalues.
>
> ---
>
> **M2**. The paper can be quite wordy, and repeat many of the same observations
>
> We will review the manuscript to ensure that the repetitions are necessary for the overall coherence of the paper.

---

> ### Author Response · Authors · 2023-11-17
> **Response to Reviewer w5ts (2/2)**
>
> **M3**. A detailed explanation of computational complexity
>
> To incorporate your comment, we provide extensive investigation on the increase in computational complexity incurred by our idea.
>
> Table 1 - Theoretical & actual time complexity for preprocessing ($b$: branching factor, $k$: number of layers, and $d_{avg}$: average degree of nodes)
>
> |                             | PathNN-SP             | DRew              | RFGNN                      | NBA-GNN                         |
> |-----------------------------|-----------------------|-------------------|----------------------------|---------------------------------|
> | Theoretical time complexity |$ \|V\| * b^k$           | $\|E\|\sim\|V\|*\|E\|$ | $\|V\|! / \|V-k-1\|!$        | $d_{avg}\|E\|$                          |
> | Actual time                 | 666s                  | 123s              | N/A                        | 68s                             |
> | Content                     | singles shortest path | $k$-hop indices     | $k$-depth non-redundant tree | Non-backtracking edge adjacency |
>
>
> Table 2 - Average test time per epoch (Tested on a single RTX 3090, the batch size inside the parentheses)
>
> | Model          | peptides-func | peptides-struct | VOC-SP     |
> |----------------|---------------|-----------------|------------|
> | PathNN-SP      | 3.349 (64)    | 2.120 (64)      | N/A        |
> | PathNN-SP+     | 5.156 (64)    | 1.417 (64)      | N/A        |
> | PathNN-AP      | 5.977 (64)    | 1.372 (64)      | N/A        |
> | GraphGPS+LapPE | 0.639 (128)   | 0.926 (128)     | 1.307 (32) |
> | DRew-GCN+LapPE | 0.933 (128)   | 1.053 (128)     | 6.468 (20) |
> | NBA-GCN+LapPE  | 0.541 (200)   | 0.532 (200)     | 5.866 (30) |
>
> One can observe that our NBA-GNN indeed increases the computational cost, but our method still achieves superior time complexity over previous works that alleviate over-squashing.

---

### Author Response · Authors · 2023-11-17
**General Response (1/2)**

# General Response

We thank you for your thoughtful feedback! We have addressed the comments and updated our draft accordingly. **All reviewers** find the experiments convincing, demonstrating the superiority of NBA-GNNs in the given tasks. The paper is praised for its clarity ($\textcolor{violet}{w5ts}$, $\textcolor{blue}{Q7oz}$)  thorough analysis of message flow redundancy, and well-motivated approach using non-backtracking updates ($\textcolor{red}{WPCH}$). The theoretical results contribute to understanding ($\textcolor{blue}{Q7oz}$, $\textcolor{green}{ayH6}$), and the use of a well-established concept from other fields to enhance GNNs is appreciated ($\textcolor{lime}{apAV}$).

Given the common viewpoints among reviewers, we will first **describe our difference from the related work LGNN in 1**, and **concerns on our theoretical analysis in 2 and 3** as follows:

---

## 1. Lack of comparison with Line Graph Neural Network

We thank the reviewers for pointing out our lack of comparison with a significant related work: line graph neural network (LGNN, Chen et al., 2019). We fully agree that the comparison is crucial and provide the conceptual and empirical comparison in what follows. We also incorporated our comparisons in the updated manuscript, and highlighted them in blue.

As the reviewers mentioned, LGNN was the first to consider the non-backtracking operator in GNNs. In comparison, **our main contribution over LGNN is in consideration of the non-backtracking operator specifically for the message redundancy problem**. Indeed, LGNN does not alleviate the redundancy problem due to contaminating the non-backtracking update with a conventional GNN update (that may backtrack).

We further support our claim that **LGNN fails to solve the message redundancy problem by empirically comparing** NBA-GNN with LGNN on the long-range graph benchmark (peptides-func), as reviewer $\textcolor{blue}{Q7oz}$ suggested. In the below table, one can observe how LGNN fails to solve the long-range problem.

Table 1 - Comparison of LGNN, NBA-GCN in peptides-func dataset
| Peptides-func | best training AP  | best test AP | Test time per epoch | RTX 3090 |
|---------------|:-----------------:|:------------:|:-------------------:|:--------:|
| LGNN          |       0.4307      |    0.4247    |     87.440 (64)     |  97.10%  |
| NBA-GCN       |       0.9724      |    0.7207    |     0.541 (200)     |  51.18%  |

Additionally, note that we have found LGNN to require prohibitively large space and time complexity for the dataset in the long-range graph benchmark due to (1) the sequential interaction between graph and line graph representation updates and (2) the use of powers of adjacency matrices, e.g., $A^2$, for message aggregation.

---

## 2. Lack of significance in Theorem 1 ($\textcolor{violet}{w5ts}$, $\textcolor{blue}{Q7oz}$, $\textcolor{green}{ayH6}$)

We resonate with the reviewers that our upper bound in Theorem 1 is not necessarily tight and derived using rather simple techniques. Nevertheless, we like to emphasize that **our contribution is the first to compare the degree of over-squashing between GNNs, aligning with different random walks**. Furthermore, demonstrating the tightness of sensitivity bounds requires unrealistic strong assumptions, e.g., there always exist parameters that set the sensitivity term to zero.

---

> ### Author Response · Authors · 2023-11-17
> **General Response (2/2)**
>
> ## 3. Lack of novelty in Theorems 2, 3 ($\textcolor{violet}{w5ts}$, $\textcolor{green}{ayH6}$, $\textcolor{lime}{apAV}$)
>
> In Theorems 2 and 3, we demonstrate the expressive capabilities of NBA-GNNs.
> Previous research has shown that the spectrum of the non-backtracking matrix possesses a valuable property for revealing hidden structures, even in very sparse settings. Specifically, [1] demonstrates the non-backtracking matrix's spectral separation property and the presence of an eigenvector containing information about the community index of vertices. Additionally, [2] reveals that analyzing the distribution of eigenvalues allows us to distinguish whether a graph originates from the Erdős–Rényi model or the SBM.
>
> Building on these foundational works, we establish the existence of a function of the spectrum of the non-backtracking matrix that can accurately recover its hidden structure. We also demonstrate that a series of convolutional layers in NBA-GNNs can act as this function by approximating the spectrum of the non-backtracking matrix. This shows the theoretical evidence of enhanced expressive power of NBA-GNNs in both node and graph classification tasks.
>
> While this approach may not seem entirely novel, it contributes to a deeper understanding of the expressive power within GNN structures by establishing connections between prior research on the non-backtracking matrix and GNNs.
>
> [1] Ludovic Stephan and Laurent Massoulié. Non-backtracking spectra of weighted inhomogeneous random graphs. Mathematical Statistics and Learning, 5(3):201–271, 2022.
>
> [2] Charles Bordenave, Marc Lelarge, and Laurent Massoulié. Non-backtracking spectrum of random graphs: community detection and non-regular Ramanujan graphs. In 2015 IEEE 56th Annual Symposium on Foundations of Computer Science, pp. 1347–1357. IEEE, 2015.

---

### Meta-Review · Area_Chair_EZJn · 2023-12-05

**Metareview:**

Important references were missed in the original submission, as pointed out by several referees. In view of these references, the novelty of the present submission seems limited and not sufficient for acceptance.

**Justification For Why Not Higher Score:**

The non-backtracking idea was explored in GNNs in the past, the paper does not add significant new insight or significant improvement over SOTA.

**Justification For Why Not Lower Score:**

N/A

---

### Decision · Program_Chairs · 2024-01-16

Reject